# Efficient and Quantization-Friendly Symbolic Fourier Convolution Algorithms

## Abstract

Fast convolution algorithms like Winograd and the Fourier transform are well-known for their substantial reduction in the multiplication complexity of Convolutional Neural Networks. However, when these methods are combined with model quantization, their inherently complex transformation matrices can introduce significant numerical errors, leading to severe degradation in model accuracy. Aiming to enhance model computation efficiency by combining fast convolution algorithms and model quantization, we introduce a novel fast convolution algorithm. This algorithm utilizes ternary matrices, with coefficients limited to ±1 and 0, for input and weight transformations, ensuring compatibility with quantization. Derived from the implementation of symbolic arithmetic on the Fourier transform, we eliminate the involvement of irrational numbers in algorithms. Further, we incorporate correction terms to convert ineffective circular convolution results into efficient ones to enhance algorithm efficiency. Additionally, we propose a corresponding post-training quantization method that requires only a few samples for calibrating network parameters and restoring accuracy without the heavy cost of retraining. Our algorithms achieve 3.68×, 4.89×, and 4.54× theoretical multiplication complexity reduction for 3×3, 5×5, and 7×7 convolutions, respectively. For models trained on the ImageNet dataset, our algorithms with the post-training method, demonstrate an accuracy drop of less than 0.2% and a reduction in bit-operations of 1.71× to 3.09× compared to Int8 quantization alone, surpassing other approaches with similar computation efficiency.

## 1 Introduction

Convolutional Neural Networks (CNNs) have achieved remarkable performance across various computer vision tasks. However, their substantial computational demands limit their deployment on edge devices (Russakovsky et al., 2015; Redmon & Farhadi, 2018; Liu et al., 2022). Quantization and Fast Convolution Algorithms are two distinct approaches to mitigate the computational burden of CNNs. Quantization methods reduce the cost of a single arithmetic operation and data transmission by converting floating-point representations to fixed-bit-width integer representations. Whereas, fast convolution algorithms reduce the number of multiplications in convolutions by adopting an equivalent computational paradigm typically consisting of three stages: transformations of weights and inputs, element-wise multiplication, and transformation for generating outputs.

However, Quantization and Fast Convolution Algorithms are not orthogonal and cannot be combined at will without negative consequences. When combining the two methods in the expectation of achieving higher computational efficiency, the introduced numerical errors are much larger than when only one method is used, potentially leading to severe model accuracy degradation. For example, Winograd is a well-known fast convolution algorithm for small filter sizes (Lavin & Gray, 2016), but its transformation matrix is ill-conditioned (with a high matrix condition number), necessitating the use of high dynamic range arithmetic to avoid numerical issues (Barabasz et al., 2020). Another renowned algorithm for accelerating convolution is the Fourier transform. While its transformation is well-conditioned, its irrational coefficients can introduce rounding errors, which is unfriendly for low-precision quantization. Additionally, its multiplication complexity exceeds that of Winograd.

At present, two research approaches have been delved to tackle the aforementioned challenge. One approach involves customizing the quantization method specifically optimized for fast convolution

algorithms (Chikin & Kryzhanovskiy, 2022; Andri et al., 2022; Li et al., 2021). However, this approach struggles to maintain satisfactory accuracy under In8 quantization for faster algorithms such as Winograd F(4×4, 3×3). The other approach is to explore new fast convolution algorithms that are better suited for quantization (Liu & Mattina, 2020; Alam et al., 2022). Nevertheless, these emerging algorithms encounter challenges in achieving low theoretical computational complexity. In summary, achieving low computational complexity, a low quantization bit-width, and preserving model accuracy simultaneously remains a challenge.

In this paper, we aim to formulate a novel fast convolution algorithm characterized by both low computational complexity and minimal quantization errors, overcoming the aforementioned challenge and achieving further computation efficiency. We leverage the numerical stability of the Fourier transform and employ symbolic computation to calculate transformations under polynomial representation, thus mitigating rounding errors originating from irrational coefficients. We refer to the proposed algorithms as "Symbolic Fourier Convolution" due to its fundamental principle, and their transformation matrices for filters and inputs contain only 1, -1, and 0. Recognizing that the conventional Fourier method utilizes only a subset of the circular convolution results, we introduce additional calculations to convert discarded terms into usable ones, thereby enhancing algorithm efficiency. Moreover, we complement the proposed algorithm with a Post-Training quantization method to achieve Int8 arithmetic while maintaining model accuracy.

In summary, our key contributions are:

1. We formulate a quantization-friendly Symbolic Fourier Convolution (SFC) algorithm for CNN accelerating. This algorithm exclusively employs coefficients of ±1 and 0 in the transformation for both filters and inputs, minimizing the adverse impacts on subsequent quantization processes. Additionally, we introduce an algorithm adjustment method that incorporates additional calculations to enhance algorithm efficiency and enable the customization of input tile size while preserving the core structure of the transformation matrix.

2. We propose a corresponding post-training method to get the quantized fast model. Our observation reveals a strong correlation between the energy distribution in the frequency domain and the frequency channel coordinates, that energy in lower-frequency tends to be higher than energy higher-frequency. Therefore, we adopt a frequency-wise strategy to calibrate the quantization scaling factor.

3. Theoretical arithmetic reduction achieved by the Symbolic Fourier Convolution (SFC) algorithm reaches up to 3.68×, 4.89×, and 4.54× for 3×3, 5×5, and 7×7 filters, respectively. Experimental results on the ImageNet dataset validate that SFC with post-training quantization method effectively maintains model accuracy at Int8 with less than a 0.2% accuracy drop, surpassing significantly comparable approaches with similar computation efficiency.

## 2 RELATED WORK

Fourier transform was the first utilized fast algorithm (Mathieu et al., 2014) to reduce the computational complexity of training Convolutional Neural Networks (CNNs). Subsequently, for small convolutions, the Winograd minimum filtering algorithm (Lavin & Gray, 2016) was found that outperformed the Fourier-based method due to its real-domain arithmetic operations, whereas the Fourier method requires more inefficient complex-domain arithmetic. Additionally, the Number Theoretic Transform (NTT) has also been proposed to accelerate convolutions (Hong et al., 2022). However, when combining Quantization and Fast Convolution Algorithms, there arises the challenge of potential model accuracy degradation. The Winograd algorithm is susceptible to numerical instability due to the ill-conditioned Vandermonde matrix in the transformation (Vincent et al., 2017; Barabasz et al., 2020). Fourier-based methods demand a high precision format to accurately represent irrational numbers. NTT methods can offer accurate integer computing, but involve a large number of modulo operations, reducing computation efficiency.

Some approaches attempt to optimize the quantization method to regain model accuracy. LoWino (Li et al., 2021) present a post-training quantization (PTQ) method for Winograd, optimizing the scaling factor by minimizing the KL distance between the quantized and original vectors. Another PTQ work (Chikin & Kryzhanovskiy, 2022) introduces a balance operation between the filter and

input channels to enhance bit-width utilization and improve the quality of quantization for Winograd. Additionally, a quantization-aware training(QAT) method with tap-wise scaling scheme as been proposed (Andri et al., 2022), which successfully restores model accuracy when employing the Wino($4\times4$, $3\times3$) algorithm with Int8 input/filter and Int10 intermediate data. Nevertheless, the above methods often struggle to achieve satisfactory accuracy recovery under Int8 quantization.

Another approach focuses on enhancing the numerical stability of the fast algorithm itself. A bilinear approach has been proposed (Barabasz & Gregg, 2019) that strikes a balance between computational complexity and numerical accuracy. Two existing works (Barabasz et al., 2020; Alam et al., 2022) aimed to discover more effective polynomial root points to improve numerical accuracy. The Winograd algorithms have also been extended to the Residue Number System (RNS) (Liu & Mattina, 2020) , decomposing single high-precision intermediate multiplications into multiple low-precision arithmetics (e.g., 8-bit), however it comes at the cost of increased computational complexity.

## 3   SYMBOLIC FOURIER CONVOLUTION ALGORITHM

Fast convolution algorithms like Winograd, Fourier transform, and NTT all employ a three-stage computing process: transformations of filters and inputs, element-wise multiplication, and a transformation for generating outputs. The generalized form for 2D convolution is as follows:

$$y = A^T[[GfG^T] \odot [BxB^T]]$$ (1)

$\odot$ denotes element-wise multiplication, and $G$, $B$ and $A$ represent the linear transformations of the input, filter, and multiplication result. The distinction in the actual computation process among these algorithms primarily resides in the number domain they operate in. Winograd algorithms work within the real domain, whereas FFT methods function within the complex domain, and NTT methods operate within the finite domain. The order of one-time arithmetic overhead is Winograd < FFT < NTT, but in terms of numerical instability, the order is Winograd > FFT > NTT.

While it's worth noting that the conventional Fourier transform exhibits superior numerical stability when dealing with large filter convolutions, we believe it holds the potential to be adapted for low-precision quantization. Nevertheless, we face two formidable challenges: the presence of irrational Fourier coefficients, which result in pronounced rounding errors during low-bit quantization, and the notably lower efficiency of the Fourier method compared to the Winograd. We address these challenges through two key improvements. Firstly, we employ symbolic computation, rather than numerical computation, to implement the discrete Fourier transform (DFT). We also represent these computational steps with matrix operation forms, such as $GfG^T$ and $B^TxB$. Subsequently, we carefully select the optimal transformation point number and introduce correction terms into the matrix operations to fully exploit the cyclic convolution output generated by the Fourier method, thereby enhancing the efficiency of our algorithms.

*1) Fourier convolution over symbolic computation*

Generally, the coefficients of the N-point DFT are derived from:

$$e^{\frac{2\pi n}{N}j} = cos(\frac{2\pi n}{N}) + jsin(\frac{2\pi n}{N}), n = 0, 1, .., N-1$$

when $\frac{n}{N} \notin \{0, \frac{1}{4}, \frac{1}{4}, \frac{3}{4}\}$, irrational values will introduce. To eliminate the rounding errors arising from these irrational values, we employ symbolic computation for the DFT rather than numerical methods. This approach represents irrational values in polynomial form with integer coefficient.

To illustrate, we consider the 3-point DFT. For a real input sequence $x = (x_0, x_1, x_2)^T$, the DFT result can be calculated as follows:

$$\begin{bmatrix} X_0 \\ X_1 \\ X_2 \end{bmatrix} = \begin{bmatrix} 1 & 1 & 1 \\ 1 & s & s^2 \\ 1 & s^2 & s \end{bmatrix} \begin{bmatrix} x_0 \\ x_1 \\ x_2 \end{bmatrix}, s = -e^{\frac{2\pi}{3}j}$$ (2)

We do not substitute the numerical value of s into the calculation. Instead, we represent and compute subsequent variables using the polynomial form of s. This allows us to express the DFT outputs $X_n$ as $X_n = X_{n,0} + X_{n,1} \cdot s + X_{n,2} \cdot s^2$.

By exploiting the geometric symmetry between $1, s, s^2$ ,the 2nd-order term of $s$ can be expressed by the opposite of the sum of 1 and s, which can reduce the number of unique components by one-third.

$$s^2 = -e^{\frac{4\pi}{3}j} = -\frac{1}{2} + \frac{\sqrt{3}}{2}j = -(1 - \frac{1}{2} - \frac{\sqrt{3}}{2}j) = -(1 + s) \tag{3}$$

$$X_n = X_{n,0} + X_{n,1} \cdot s + X_{n,2} \cdot s^2 = X_{n,0} - X_{n,2} + (X_{n,1} - X_{n,2})s = X'_{n,0} + X'_{n,1}s \tag{4}$$

Further, the Hamiltonian symmetry of the real signal Fourier transform can reduce the number of components by almost half. Thus, the symbolic computational form of the Fourier transform can be expressed as following:

$$\begin{bmatrix} X_0 \\ X_1 \\ X_2 \end{bmatrix} = \begin{bmatrix} X'_{0,0} \\ X'_{1,0} + sX'_{1,1} \\ X'_{1,0} - sX'_{1,1} \end{bmatrix}, \text{where } \begin{bmatrix} X'_{0,0} \\ X'_{1,0} \\ X'_{1,1} \end{bmatrix} = \begin{bmatrix} 1 & 1 & 1 \\ 1 & 0 & -1 \\ 0 & 1 & -1 \end{bmatrix} \begin{bmatrix} x_0 \\ x_1 \\ x_2 \end{bmatrix} \tag{5}$$

In the above formula, $s = -e^{\frac{2\pi}{3}j}$ does not need to be explicitly included in the calculation but serves as a notation from the outset. Similarly, the multiplication in the frequency domain needs to be redefined.

$$(a_0 + a_1 s) * (b_0 + b_1 s) = a_0 b_0 - a_1 b_1 + (a_0 b_1 + a_1 b_0 - a_1 b_1)s = e - f + (g + e)s,$$
$$\text{where } e = a_0 b_0, f = a_1 b_1, g = (a_1 - a_0)(b_0 - b_1) \tag{6}$$

The multiplication of two 1st-order polynomials can be seen as the convolution of two sequences of length 2. Therefore we can perform minimal 3 real multiplications to multiply $(a_0 + a_1 s)$ and $(b_0 + b_1 s)$.

Let's delve into the general N-point real signal DFT. For symbolic computation, we need $N$ Nth-order polynomials. However, the number of these polynomials can be reduced by more than half when we take advantage of Hermitian symmetry. Additionally, the order of the polynomials can be decreased through geometric symmetry. Specifically, if $N$ has prime factors $m_0, m_1, .., m_{L-1}$, its polynomial order can be reduced to $\prod_{l=0}^{L-1} \frac{m_l - 1}{m_l}$. For instance, there's a symmetry law $s^{N-j} = -s^j$ for $N$ that contains factors of 2, effectively halving the polynomial order. Furthermore, when N contains a factor of 3, a symmetry $s^{N-j} = -s^j - 1$ exists, leading to a reduction in the polynomial order by one-third. Multiplying two polynomials of order $O$ can be seen as the convolution of two sequences of length $O$ and requires a minimum of $2O - 1$ real multiplications. Thus, we can estimate the multiplicative efficiency under symbolic computation.

Through enumeration, we can identify that both 6 and 4 are suitable choices for the number of DFT points in fast convolution applications. This is because they exhibit relatively lower computational demands and can effectively accommodate commonly used 3×3 filters.

Considering DFT-6, its transformation coefficients consist of six values: $1, e^{j\frac{\pi}{3}}, e^{j\frac{2\pi}{3}}, -1, e^{j\frac{4\pi}{3}}, e^{j\frac{5\pi}{3}}$. Let's define $s = e^{\frac{\pi}{3}j}$, and then it follows that $s^2 = s - 1$, allowing all coefficients to be expressed as first-order polynomials of $s$: $1, s, s - 1, -1, -s, 1 - s$. When multiplying two first-degree polynomials, any quadratic term can be reduced to a first-degree term using the rule $s^2 = s - 1$. Therefore, the DFT-6 transform processing under symbolic computation is as follows:

$$DFT6(x) = S_6 F_6 \boldsymbol{x} = \begin{bmatrix} 1 & 0 & 0 & 0 & 0 & 0 \\ 0 & 1 & s & 0 & 0 & 0 \\ 0 & 0 & 0 & 1 & s & 0 \\ 0 & 0 & 0 & 0 & 0 & 1 \\ 0 & 1 & -s & 0 & 0 & 0 \\ 0 & 0 & 0 & 1 & -s & 0 \end{bmatrix} \begin{bmatrix} 1 & 1 & 1 & 1 & 1 & 1 \\ 1 & 1 & 0 & -1 & -1 & 0 \\ 0 & -1 & -1 & 0 & 1 & 1 \\ 1 & 0 & -1 & 1 & 0 & -1 \\ 0 & -1 & 1 & 0 & -1 & 1 \\ 1 & -1 & 1 & -1 & 1 & -1 \end{bmatrix} \begin{bmatrix} x_0 \\ x_1 \\ x_2 \\ x_3 \\ x_4 \\ x_5 \end{bmatrix} \tag{7}$$

Here, $S_6$ represents the transition from symbolic to numerical computation without any arithmetic, and $T_6$ signifies the Fourier transform under symbolic computing. We refer to the intermediate matrix as SFT-6 (Symbolic Fourier Transform-6), as its coefficients consist solely of 1, -1, and 0.

Similarly, the DFT-4 under symbolic computing can be constructed in the same manner.

$$DFT4(x) = S_4 F_4 \boldsymbol{x} = \begin{bmatrix} 1 & 0 & 0 & 0 \\ 0 & 1 & j & 0 \\ 0 & 0 & 0 & 1 \\ 0 & 1 & -j & 0 \end{bmatrix} \begin{bmatrix} 1 & 1 & 1 & 1 \\ 1 & 0 & -1 & 1 \\ 0 & -1 & 0 & 1 \\ 1 & -1 & 1 & -1 \end{bmatrix} \begin{bmatrix} x_0 \\ x_1 \\ x_2 \\ x_3 \end{bmatrix} \tag{8}$$

In the element-wise multiplication steps, multiplications are performed in polynomial form. Multiplying two 1st-order polynomials requires 4 real number multiplications. To reduce this cost, we can utilize a short fast convolution algorithm. The 2nd-order term of the resulting 2nd-degree polynomial must be combined with the 0th-order and 1st-order terms. By employing the fast algorithm, we can calculate one 1st-order polynomial multiplication with just 3 real multiplications.

For DFT-6:

$$(a_0 + a_1 s) * (w_0 + w_1 s) = \begin{bmatrix} 1 & -1 & 0 \\ -1 & 0 & 1 \end{bmatrix} (\begin{bmatrix} 1 & 0 \\ 0 & 1 \\ 1 & 1 \end{bmatrix} \begin{bmatrix} a_0 \\ a_1 \end{bmatrix} \odot \begin{bmatrix} 1 & 0 \\ 0 & 1 \\ 1 & 1 \end{bmatrix} \begin{bmatrix} w_0 \\ w_1 \end{bmatrix}) \tag{9}$$

For DFT-4:

$$(a_0 + a_1 j) * (w_0 + w_1 j) = \begin{bmatrix} 1 & 1 & 0 \\ -1 & -1 & 1 \end{bmatrix} (\begin{bmatrix} 1 & 0 \\ 0 & 1 \\ 1 & 1 \end{bmatrix} \begin{bmatrix} a_0 \\ a_1 \end{bmatrix} \odot \begin{bmatrix} 1 & 0 \\ 0 & 1 \\ 1 & 1 \end{bmatrix} \begin{bmatrix} w_0 \\ w_1 \end{bmatrix}) \tag{10}$$

If we wish to compute $A((Gf) \odot (Bx))$ directly in the real number domain, akin to the Winograd algorithm, without involving polynomial multiplication, we can integrate Eq.(9) or Eq.(10) into the SFT matrix, as shown in Eq.(7) or Eq.(8). In the 1D case, this does not impact efficiency. However, in the 2D case, it introduces redundant components and marginally reduces the acceleration ratio.

*2) Adding correction terms to achieve linear convolution and higher efficiency*

The Fourier method can directly produce circular convolution. However, in conventional practice, to achieve linear convolution with an $r \times r$ filter size, only $(n - r + 1) \times (n - r + 1)$ elements of the $n \times n$ cyclic convolution are valid, while the rest are discarded as waste. This waste is another crucial factor affecting the efficiency of the FFT method, despite its complex arithmetic. Therefore, we aim to make use of this waste by introducing a modification operation.

Fig.1 illustrates an example of Fourier-based cyclic convolution for $n = 6$ and $r = 3$. The fisrt term CyclicO1 is equal to $a_6 w_1 + a_1 w_2 + a_2 w_1$, but the desired output is LinearO1 $= a_0 w_1 + a_1 w_2 + a_2 w_1$. To align LinearO1 with CyclicO1, we introduce a corrective term, obtain the desired output LinearO1 $=$ CyclicO1 $+ (a_6 - a_0) w_1$. With this adjustment, adding just one MAC operation allows us to obtain an additional correct result, utilizing the Fourier convolution output more efficiently compared to the previous approach of discarding erroneous terms. Note that while matrix A may not be as straightforward as matrices G and B, it operates on the data obtained after multiplication with a larger bit-width, hence it does not introduce any negative effects.

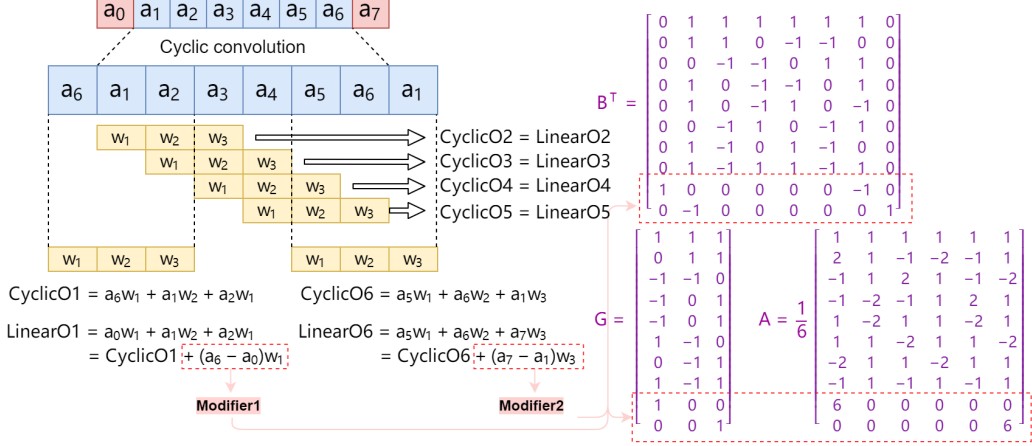

Figure 1: Converting cyclic convolution to Linear convolution.

To unambiguously represent a particular algorithm, we employ the notation SFC-$n(r, k)$, where $n$ signifies the length of the SFT transformation, $r$ denotes the feature tile size, and $k$ represents the

kernel size. For example, the SFC-6(6×6,3×3) algorithm is constructed based on a 6-point Fourier transform, employing a 3×3 kernel size, and utilizing a 6×6 feature tile size.

By introducing correction terms, we can also adapt SFC for different input tile sizes. For example, when calculating the convolution based SFT-6 with a tile size of 7. It's worth noting that the images in the ImageNet dataset have an original size of 224×224, which is a multiple of 7. Utilizing the SFC-6(7×7, 3×3) algorithm for processing networks designed for Imagenet would result in higher tiling efficiency without the need for padding. The transformation matrix integrated polynomial multiplication of the SFC-6(7×7, 3×3) is as follows:

$$
B^T = \begin{bmatrix}
0 & 1 & 1 & 1 & 1 & 1 & 1 & 0 & 0 \\
0 & 1 & 1 & 0 & -1 & -1 & 0 & 0 & 0 \\
0 & 0 & -1 & -1 & 0 & 1 & 1 & 0 & 0 \\
0 & 1 & 0 & -1 & -1 & 0 & 1 & 0 & 0 \\
0 & 1 & 0 & -1 & 1 & 0 & -1 & 0 & 0 \\
0 & 0 & -1 & 1 & 0 & -1 & 1 & 0 & 0 \\
0 & 1 & -1 & 0 & 1 & -1 & 0 & 0 & 0 \\
0 & 1 & -1 & 1 & -1 & 1 & -1 & 0 & 0 \\
1 & 0 & 0 & 0 & 0 & 0 & -1 & 0 & 0 \\
0 & -1 & 0 & 0 & 0 & 0 & 0 & 1 & 0 \\
0 & -1 & 0 & 0 & 0 & 0 & 0 & 1 & 0 \\
0 & 0 & -1 & 0 & 0 & 0 & 0 & 0 & 1
\end{bmatrix},
$$

$$
G = \begin{bmatrix}
1 & 1 & 1 \\
0 & 1 & 1 \\
-1 & -1 & 0 \\
-1 & 0 & 1 \\
-1 & 0 & 1 \\
1 & -1 & 0 \\
0 & -1 & 1 \\
1 & -1 & 1 \\
1 & 0 & 0 \\
0 & 0 & 1 \\
0 & 1 & 0 \\
0 & 0 & 1
\end{bmatrix}, A = \frac{1}{6} \begin{bmatrix}
1 & 1 & 1 & 1 & 1 & 1 \\
2 & 1 & -1 & -2 & -1 & 1 \\
-1 & 1 & 1 & 1 & -1 & -2 \\
-1 & -2 & -1 & 1 & 2 & 1 \\
1 & -2 & 1 & 1 & -2 & 1 \\
1 & 1 & -2 & 1 & 1 & -2 \\
-2 & 1 & 2 & -2 & 1 & 1 \\
-1 & 1 & -1 & 1 & -1 & 1 \\
6 & 0 & 0 & 0 & 0 & 0 \\
0 & 0 & 0 & 0 & 6 & 0 \\
0 & 0 & 0 & 0 & 0 & 6 \\
0 & 0 & 0 & 0 & 0 & 6
\end{bmatrix}
$$

(11)

The SFC-6(6×6, 3×3) algorithm can reduce 73% of the multiplications in 3×3 convolutions. Similarly, leveraging the SFT-6 core algorithm, 5×5 and 7×7 convolutions can optimize 81% and 79% of multiplications, respectively. A selection of achievable Symbolic Fourier Convolution algorithms is listed in Table 1. In addition to comparing the reduction ratios in multiplication complexity, we evaluate the numerical errors introduced by different algorithms. Using the results computed in FP32 by the direct convolution as the reference, we calculate the fp16 results of various fast algorithms and determine the average Mean Squared Error (MSE) between them. For a kernel size of 3×3, both the Winograd F(4×4, 3×3) algorithm and the SFC-6(6×6, 3×3) algorithm exhibit roughly equal relative multiplication complexity. However, the SFC algorithm shows only half of the average error compared to the former. For larger kernel sizes, the SFC algorithm demonstrates a more significant advantage in terms of numerical error.

Table 1: Comparison of Fast Convolution Algorithms.

| Algorithm | Kernel Size | Tile Size | Normalized Error | Related Mult. Complexity |
|---|---|---|---|---|
| Wino(3×3, 3×3) | 3×3 | 3×3 | 3.4 | 30.4% |
| SFC-4(4×4, 3×3) | 3×3 | 4×4 | 1.9 | 31.94%/34.03% |
| Wino(4×4, 3×3) | 3×3 | 4×4 | 4.8 | 25% |
| SFC-6(6×6, 3×3) | 3×3 | 6×6 | 2.2 | 27.16%/30.87% |
| SFC-6(7×7, 3×3) | 3×3 | 7×7 | 2.3 | 29.93%/32.65% |
| Wino(2×2, 5×5) | 5×5 | 2×2 | 9.5 | 36% |
| SFC-6(6×6, 5×5) | 5×5 | 6×6 | 2.1 | 20.44%/21.78% |
| Wino(2×2, 7×7) | 7×7 | 2×2 | 18.0 | 32.6% |
| SFC-6(4×4, 7×7) | 7×7 | 4×4 | 2.7 | 21.99%/25% |

## 4 POST-TRAINING QUANTIZATION FOR SFC

In this section, we will introduce the tailored Post-training Quantization (PTQ) method for SFC. PTQ is a low computational cost quantization after the model has been trained. It involves converting the model's high-precision parameters (usually 32-bit floating-point) into lower-precision

representations (such as 8-bit integers) without retraining the model. The PTQ method for starndard CNNs can achieve nearly lossless accucarcy compared to floating model under 8-bit quantization. However, when it comes to the Winograd CNNs, requires the use of a more computationally expensive Quantization-Aware-Training(QAT) method to achieve similar accuracy at 8-bit. This is due to the negative effects caused by the complex transformation coefficients. Even though the SFC algorithm has extremely simple transformation coefficients, we still need to consider the potential overflow caused by data accumulation. Therefore, in the PTQ (Post-Training Quantization) scheme, we incorporate three techniques - frequency channel quantization, knowledge distillation, and quantization scale-factor fine-tuning to ensure effectiveness.

## 4.1 FREQUENCY-WISE QUANTIZATION

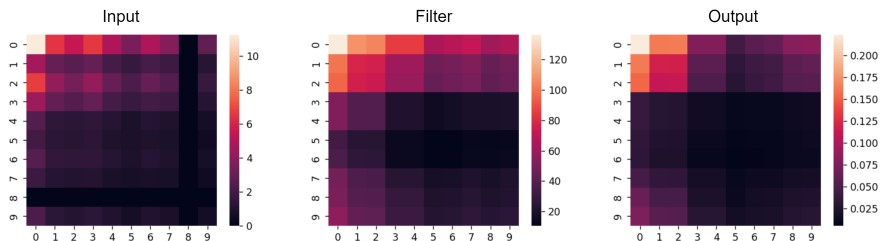

Figure 2: The frequency domain energy distribution of the 9-th conv-layer in Resnet-18 on ImageNet

Since the SFC algorithm is the expansion of the Fourier transform, considering the distribution of image data in the frequency domain is highly related to the frequency channel index, as Fig.2 shows, we adopt a frequency-wise quantization approach, which can be represented by the following equation:

$$y = \sum_{C_{in}} (s_{Tx} \left\lceil B^T x B / s_{Tx} \right\rfloor_{int N_{Tx}} \odot s_{Tf} \left\lceil G f G^T / s_{Tf} \right\rfloor_{int N_{Tf}}) \tag{12}$$

The scaling factor $s_{Tx}$ is in size $[T \times T]$, where $T$ is the size of the transform domain. For the quantization factor $s_{Tf}$ of weights, considering that per-channel quantization can achieve better results in regular convolutions, we suggest combine per-frequency quantization and per-channel quantization that the $s_{Tf}$ is in size $[OC \times T \times T]$ to achieve higher accuracy.

## 4.2 POST-TRAINING QUANTIZATION FOR SFC

We employ a pre-trained FP32 model as the teacher to fine-tune the weights and scale factors of the quantized model. Approximately one hundred unlabeled data samples were randomly selected and fed into both the floating-point model and the quantized model to obtain intermediate layer features from each convolution layer's output. The Mean Squared Error (MSE) distance between the output features generated by the FP32 and quantized layers serves as the loss function for adjusting the weights and scale factors. We utilize the straight-through estimator to compute backpropagation gradients for the weights, and the backpropagation method for scale factors is elaborated on in Section 4.3. The formula for knowledge distillation is as follows:

$$\underset{w,s}{argmin} \ \|(L_{FP32}(x), L_{int_N}(x, w, s))\|_F^2 \tag{13}$$

## 4.3 SCALING FACTOR FINE-TUNE

The initial scale factors are determined based on the maximum and minimum values within the floating-point data distribution. To mitigate the impact of rounding and truncation errors and enhance quantization performance, we employ scaling factor fine-tuning. Following the methodology outlined in (Jain et al., 2020), we implement the backward propagation of scaling factor gradients

using the following formula:

$$\nabla_{(log_2 t)} q(x; s) := s \ln(2) \cdot \begin{cases} -2^{n-1} & \text{if } \lfloor \frac{x}{s} \rceil < -2^{n-1}, \\ 2^{n-1} - 1 & \text{if } \lfloor \frac{x}{s} \rceil > 2^{n-1} - 1, \\ \lfloor \frac{x}{s} \rceil - \frac{x}{s} & else \end{cases} \quad (14)$$

where $s = 2^{\lceil t \rceil}$.

## 5 EXPERIMENTAL EVALUATION

### 5.1 POST-TRAINING QUANTIZATION FOR SFC

*1) Experiment setting.* We conduct experiment on the ImageNet dataset (Russakovsky et al., 2015) which comprises 1.4 million images of size 224×224×3, distributed across 1,000 classes. From the training set, we randomly selected no more than 0.1% of unlabeled images to form the calibration set for post-training quantization. Model accuracy was evaluated on the validation set. For benchmarking, we utilized pre-trained Resnet18 and Resnet50 models from Torchvision. Prior to post-training quantization, all batch normalization layers were integrated into the preceding convolution layers.

*2) Evaluation.* To assess the impact of different fast algorithms on quantization, we conducted post-training quantization on the following scenarios: 1) models with standard convolution, 2) models accelerated by Winograd algorithms, and 3) models accelerated by SFC algorithms. In all cases, 3×3 convolution layers with a stride of 1 were replaced by the corresponding fast algorithms. We also involved a effective PTQ method AdaQuant (Hubara et al., 2020) on experiment. From Int8 to Int4, data in the transform domain was quantized to the corresponding bit-width, while inputs and weights in the spatial domain were quantized to Int8. This approach ensured alignment of data with external storage, allowing the computing unit with the most computational burden to benefit from quantization.

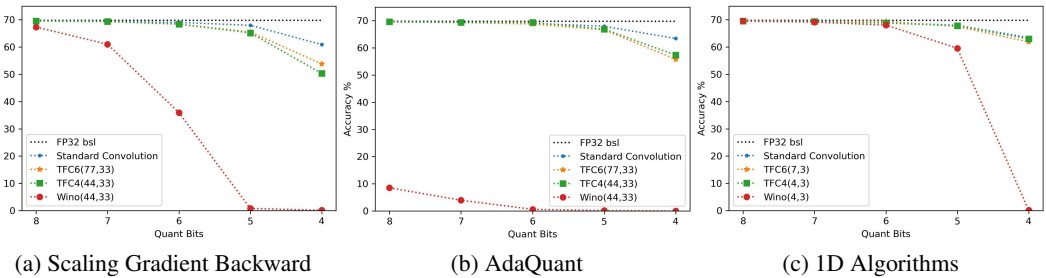

| (a) Scaling Gradient Backward | (b) AdaQuant | (c) 1D Algorithms |
|---|---|---|

Figure 3: Accuracy with respect to quantization bits of Resnet18 for different algorithms.

We plot the accuracy curves concerning quantization bits for various algorithms in Fig(4). Under Int8 quantization, we observed that SFC-4(4×4, 3×3) and SFC-6(7×7, 3×3) achieved nearly the same accuracy as quantized standard convolution, while the Winograd F(4×4, 3×3) method incurred a 2.7% accuracy loss. These results align with previous findings in Winograd's algorithm. Moving from Int7 to Int4, SFC consistently exhibited accuracy losses close to the standard method. Surprisingly, SFC-6 outperformed SFC-4 slightly, despite its larger transformation matrix introducing more accumulations and potential rounding errors. We attribute this to better alignment of SFC-6's block size with ImageNet's image dimensions, allowing for greater information preservation. Moreover, SFC-6's larger block size may share properties with the 8×8 blocks used in JPEG compression, leading to more efficient compression with minimal loss of image information. In summary, the SFC method incurred a marginal accuracy loss compared to quantized standard convolution. However, the Winograd method, constrained by its ill-conditioned transform matrices, experienced significant accuracy loss at Int8 or narrower data-widths. Winograd's accuracy declined sharply with decreasing quantization bits, rendering it impractical below 6 bits. Under Int4 quantization, AdaQuant can improve the result of standard convolution and our algorithms under 4-bit quantization. However, for the Winograd, there are convergence problems where the results are even worse. We suggested that using SFC 1D algorithms under Int4 and applying SFC 2D algorithms under Int8/Int6.

To demonstrate the practical viability of our algorithm in reducing multiplicative complexity while preserving accuracy after quantization, we conducted a comparative analysis with efficient Winograd based quantization improvement methods. These include post-training quantization (PTQ) methods such as Channel Balancing (Chikin & Kryzhanovskiy, 2022), Full Quantization (Tianqi et al., 2023), the quantization-aware-training (QAT) method Tap-wise Quantization. (Andri et al., 2022), and the Residual Numbers System (Liu & Mattina, 2020) on the ImageNet dataset.

We opt for bit-operations (BOPs) as a fine metric to precisely quantify computation efficiency. This metric comprehensively considers factors such as bit-width, operations number, and the varying hardware costs of addition and multiplication. It is widely used in various model compressing fields, including Neural Architecture Search(NAS), quantization and pruning research (Wang et al., 2020; Guo et al., 2020; Liu et al., 2020). BOPs for integer arithmetic are computed according to the following rules: for addition, they are equal to the bit-width multiplied by the number of additions; for multiplication, they are equal to the square of the bit-width multiplied by the number of multiplications.

The implementation results are presented in Table 2. Our work has achieved a significant accuracy improvement compared to state-of-the-art Winograd-based methods with similar BOPs. Notably, the SFC-6 (7×7, 3×3) algorithm demonstrates almost identical multiplication reduction capabilities to the Wino(4×4, 3×3) in Resnets deployments, thanks to its more tiling-efficient input size design. Conversely, the faster Wino(6×6, 3×3) algorithm exhibits lower practical efficiency due to its less suitable input size. This underscores the value of the SFC algorithm's adaptability in adjusting input sizes by incorporating correction terms.

Table 2: Compared with related work

| Method | Algorithm | BOPs | Bits | QuantType | Top1 | Ref. | Δ |
|---|---|---|---|---|---|---|---|
| **Resnet50** | | **216.3G** | **8** | | | | |
| Tap-wise Quant. | Wino(4×4, 3×3) | 125.1G | 8 | QAT | 75.2 | 75.5 | -0.3 |
| Channel Balancing | Wino(4×4, 3×3) | 125.1G | 8 | PTQ | 75.8 | 76.1 | -0.3 |
| Channel Balancing | Wino(6×6, 3×3)) | 129.5G | 8 | PTQ | 74.5 | 76.1 | -1.6 |
| Residual Numbers | Wino(10×10, 3×3) | 184.1G | 8 | - | 75.1 | - | - |
| Full Quant. | Wino(4×4, 3×3) | 125.1G | 8 | PTQ | 75.4 | 76.1 | -0.7 |
| Symbolic Fourier | SFC6(7×7, 3×3)) | 125.9G(1.71×) | 8 | PTQ | **76.0** | 76.1 | **-0.1** |
| **Resnet34** | | **214.6G** | **8** | | | | |
| Tap-wise Quant. | Wino(4×4, 3×3) | 68.2G | 8 | QAT | 71.1 | 72.6 | -1.5 |
| Channel Balancing | Wino(4×4, 3×3) | 68.2G | 8 | PTQ | 71.9 | 73.3 | -1.4 |
| Full Quant. | Wino(4×4, 3×3) | 68.2G | 8 | PTQ | 71.8 | 73.3 | -1.5 |
| Symbolic Fourier | SFC6(7×7,3×3) | 69.37G(3.09×) | 8 | PTQ | **73.1** | 73.3 | **-0.2** |
| **Resnet18** | | **96.2G** | **8** | | | | |
| Channel Balancing | Wino(4×4, 3×3) | 33.7G | 8 | PTQ | 67.5 | 69.7 | -2.2 |
| Channel Balancing | Wino(6×6, 3×3) | 38.6G | 8 | PTQ | 60.6 | 69.7 | -9.1 |
| Full Quant. | Wino(4×4, 3×3) | 33.7G | 8 | PTQ | 68.8 | 69.7 | -0.9 |
| Full Quant. | Wino(4×4, 3×3) | 19.3G | 8/6 | PTQ | 64.3 | 69.7 | -5.4 |
| Symbolic Fourier | SFC6(7×7, 3×3) | 34.3G(2.80×) | 8 | PTQ | **69.5** | 69.7 | **-0.2** |
| Symbolic Fourier | SFC4(4×4, 3×3) | 22.8G | **8/6** | PTQ | **68.4** | 69.7 | **-0.7** |
| Symbolic Fourier | SFC4(4, 3) | 15.8G | **8/4** | PTQ | **63.0** | 69.7 | **-6.7** |

## 6 CONCLUSION

We propose a novel fast convolution algorithm extended by Fourier transform with corresponding post-training quantization method, which solves the numerical instability problem of the conventional fast convolution algorithm (e.g. Winograd) applied to quantized CNNs. Our experiments demonstrate that it is possible to accelerate a 3×3 convolution by more than 3× at Int8 arithmetic without paying additional accuracy drop. Our algorithm can be computed in the same computational flow as the Winograd algorithm, which means that its deployment on general-purpose processors (CPUs, GPUs) and the design of hardware accelerators can follow the previous paradigm exactly.

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

# A APPENDIX

## A.1 TYPICAL SYMBOLIC FOURIER CONVOLUTION ALGORITHMS

For SFC-4($4 \times 4$, $3 \times 3$), the matrices (with polynomial multiplication integrated) are:

$$
B^T = \begin{pmatrix}
0 & 1 & 1 & 1 & 1 & 0 \\
0 & -1 & 1 & -1 & 1 & 0 \\
0 & 1 & -1 & -1 & 1 & 0 \\
0 & 0 & -1 & 0 & 1 & 0 \\
0 & 1 & 0 & -1 & 0 & 0 \\
1 & 0 & 0 & 0 & -1 & 0 \\
0 & -1 & 0 & 0 & 0 & 1
\end{pmatrix},
$$

$$
G = \begin{pmatrix}
1 & 1 & 1 \\
1 & -1 & 1 \\
1 & -1 & -1 \\
1 & 0 & -1 \\
0 & -1 & 0 \\
1 & 0 & 0 \\
0 & 0 & 1
\end{pmatrix}, A = \frac{1}{4} \begin{pmatrix}
1 & 1 & 1 & 1 \\
1 & -1 & 1 & -1 \\
0 & 2 & 0 & -2 \\
2 & -2 & -2 & 2 \\
-2 & -2 & 2 & 2 \\
4 & 0 & 0 & 0 \\
0 & 0 & 0 & 4
\end{pmatrix}
$$

(15)

It costs 49 multiplications to generate 16 outputs. And only 46 multiplications are consumed when Hermitian symmetry is fully considered.

For SFC-6($6 \times 6$, $3 \times 3$), it costs 100/88 multiplications to generate 36 outputs, the matrices are:

$$
B^T = \begin{bmatrix}
0 & 1 & 1 & 1 & 1 & 1 & 1 & 0 \\
0 & 1 & 1 & 0 & -1 & -1 & 0 & 0 \\
0 & 0 & -1 & -1 & 0 & 1 & 1 & 0 \\
0 & 1 & 0 & -1 & -1 & 0 & 1 & 0 \\
0 & 1 & 0 & -1 & 1 & 0 & -1 & 0 \\
0 & 0 & -1 & 1 & 0 & -1 & 1 & 0 \\
0 & 1 & -1 & 0 & 1 & -1 & 0 & 0 \\
0 & 1 & -1 & 1 & 1 & -1 & 1 & 0 \\
1 & 0 & 0 & 0 & 0 & 0 & -1 & 0 \\
0 & -1 & 0 & 0 & 0 & 0 & 0 & 1
\end{bmatrix},
$$

$$
G = \begin{bmatrix}
1 & 1 & 1 \\
0 & 1 & 1 \\
-1 & -1 & 0 \\
-1 & 0 & 1 \\
-1 & 0 & 1 \\
1 & -1 & 0 \\
0 & -1 & 1 \\
1 & -1 & 1 \\
1 & 0 & 0 \\
0 & 0 & 1
\end{bmatrix}, A = \frac{1}{6} \begin{bmatrix}
1 & 1 & 1 & 1 & 1 & 1 \\
2 & 1 & -1 & -2 & -1 & 1 \\
-1 & 1 & 2 & 1 & -1 & -2 \\
-1 & -2 & -1 & 1 & 2 & 1 \\
1 & -2 & 1 & 1 & -2 & 1 \\
1 & 1 & -2 & 1 & 1 & -2 \\
-2 & 1 & 1 & -2 & 1 & 1 \\
-1 & 1 & -1 & 1 & -1 & 1 \\
6 & 0 & 0 & 0 & 0 & 0 \\
0 & 0 & 0 & 0 & 0 & 6
\end{bmatrix}
$$

(16)

For SFC-6($7 \times 7$, $3 \times 3$), it costs 144/132 multiplications to generate 49 outputs, the matrices are:

$$
B^T = \begin{bmatrix}
0 & 1 & 1 & 1 & 1 & 1 & 1 & 0 & 0 \\
0 & 1 & 1 & 0 & -1 & -1 & 0 & 0 & 0 \\
0 & 0 & -1 & -1 & 0 & 1 & 1 & 0 & 0 \\
0 & 1 & 0 & -1 & -1 & 0 & 1 & 0 & 0 \\
0 & 1 & 0 & -1 & 1 & 0 & -1 & 0 & 0 \\
0 & 0 & -1 & 1 & 0 & -1 & 1 & 0 & 0 \\
0 & 1 & -1 & 0 & 1 & -1 & 0 & 0 & 0 \\
0 & 1 & -1 & 1 & -1 & 1 & -1 & 0 & 0 \\
1 & 0 & 0 & 0 & 0 & 0 & -1 & 0 & 0 \\
0 & -1 & 0 & 0 & 0 & 0 & 0 & 1 & 0 \\
0 & -1 & 0 & 0 & 0 & 0 & 0 & 1 & 0 \\
0 & 0 & -1 & 0 & 0 & 0 & 0 & 0 & 1
\end{bmatrix},
$$

$$
G = \begin{bmatrix}
1 & 1 & 1 \\
0 & 1 & 1 \\
-1 & -1 & 0 \\
-1 & 0 & 1 \\
-1 & 0 & 1 \\
1 & -1 & 0 \\
0 & -1 & 1 \\
1 & -1 & 1 \\
1 & 0 & 0 \\
0 & 0 & 1 \\
0 & 1 & 0 \\
0 & 0 & 1
\end{bmatrix},\quad
A = \frac{1}{6}\begin{bmatrix}
1 & 1 & 1 & 1 & 1 & 1 & 1 \\
2 & 1 & -1 & -2 & -1 & 1 & 2 \\
-1 & 1 & 2 & 1 & -1 & -2 & -1 \\
-1 & -2 & -1 & 1 & 2 & 1 & -1 \\
1 & -2 & 1 & 1 & -2 & 1 & 1 \\
1 & 1 & -2 & 1 & 1 & -2 & 1 \\
-2 & 1 & 1 & -2 & 1 & 1 & -2 \\
-1 & 1 & -1 & 1 & -1 & 1 & -1 \\
6 & 0 & 0 & 0 & 0 & 0 & 0 \\
0 & 0 & 0 & 0 & 0 & 0 & 6 \\
0 & 0 & 0 & 0 & 0 & 0 & 6 \\
0 & 0 & 0 & 0 & 0 & 0 & 6
\end{bmatrix}
\tag{17}
$$

For SFC-6($6 \times 6$, $5 \times 5$), it costs 196/184 multiplications to generate 36 outputs, the matrices are:

$$
B^T = \begin{bmatrix}
0 & 0 & 1 & 1 & 1 & 1 & 1 & 1 & 0 & 0 \\
0 & 0 & 1 & 1 & 0 & -1 & -1 & 0 & 0 & 0 \\
0 & 0 & 0 & -1 & -1 & 0 & 1 & 1 & 0 & 0 \\
0 & 0 & 1 & 0 & -1 & -1 & 0 & 1 & 0 & 0 \\
0 & 0 & 1 & 0 & -1 & 1 & 0 & -1 & 0 & 0 \\
0 & 0 & 0 & -1 & 1 & 0 & -1 & 1 & 0 & 0 \\
0 & 0 & 1 & -1 & 0 & 1 & -1 & 0 & 0 & 0 \\
0 & 0 & 1 & -1 & 1 & -1 & 1 & -1 & 0 & 0 \\
1 & 0 & 0 & 0 & 0 & 0 & -1 & 0 & 0 & 0 \\
0 & 1 & 0 & 0 & 0 & 0 & 0 & -1 & 0 & 0 \\
0 & 1 & 0 & 0 & 0 & 0 & 0 & -1 & 0 & 0 \\
0 & 0 & -1 & 0 & 0 & 0 & 0 & 0 & 1 & 0 \\
0 & 0 & -1 & 0 & 0 & 0 & 0 & 0 & 1 & 0 \\
0 & 0 & 0 & -1 & 0 & 0 & 0 & 0 & 0 & 1
\end{bmatrix},
$$

$$
G = \begin{bmatrix}
1 & 1 & 1 & 1 & 1 \\
-1 & -1 & 0 & 1 & 1 \\
1 & 0 & -1 & -1 & 0 \\
0 & -1 & -1 & 0 & 1 \\
0 & 1 & -1 & 0 & 1 \\
-1 & 0 & 1 & -1 & 0 \\
-1 & 1 & 0 & -1 & 1 \\
1 & -1 & 1 & -1 & 1 \\
1 & 0 & 0 & 0 & 0 \\
1 & 0 & 0 & 0 & 0 \\
0 & 1 & 0 & 0 & 0 \\
0 & 0 & 0 & 1 & 0 \\
0 & 0 & 0 & 0 & 1 \\
0 & 0 & 0 & 0 & 1
\end{bmatrix},\quad
A = \frac{1}{6}\begin{bmatrix}
1 & 1 & 1 & 1 & 1 & 1 & 1 \\
1 & -1 & -2 & -1 & 1 & 2 & -1 \\
1 & 2 & 1 & -1 & -2 & -1 & 2 \\
-2 & -1 & 1 & 2 & 1 & -1 & -1 \\
-2 & 1 & 1 & -2 & 1 & 1 & 1 \\
1 & -2 & 1 & 1 & -2 & 1 & -2 \\
1 & 1 & -2 & 1 & 1 & -2 & 1 \\
1 & -1 & 1 & -1 & 1 & -1 & -1 \\
6 & 0 & 0 & 0 & 0 & 0 & 0 \\
6 & 0 & 0 & 0 & 0 & 0 & 0 \\
0 & 6 & 0 & 0 & 0 & 0 & 0 \\
0 & 0 & 0 & 0 & 0 & 6 & 0 \\
0 & 0 & 0 & 0 & 0 & 0 & 6 \\
0 & 0 & 0 & 0 & 0 & 0 & 6
\end{bmatrix}
\tag{18}
$$

## A.2 Layer-wise MSE loss comparison of different algorithms

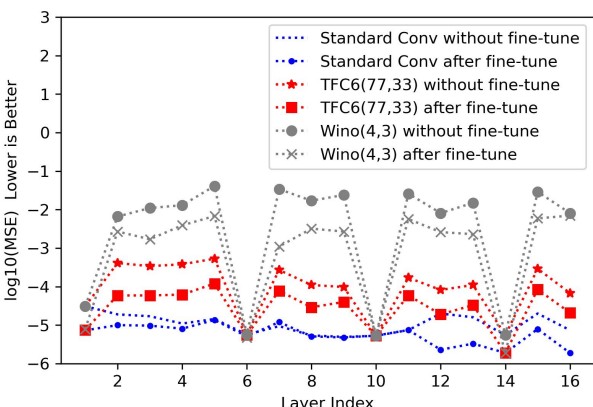

Figure 4: Layer-wise MSE loss of featuremaps in Resnet18 at Int8 quantization

We plot the layer-wise Mean Squared Error (MSE) error of feature maps as the qualification metric to compared different algorithms, using AdaQuant as the fine-tune method. We can see that, in our proposed SFC algorithm, the MSE loss of feature maps is significantly lower than that in the Winograd algorithm. However, it is worth noting that, when using the scaling-factor fine-tune method, the MSE loss of feature maps in the SFC algorithm is smaller than that in the standard convolution, while the accuracy of the quantized model does not improve. It enlightens us that the MSE error of the feature map is not the sole criterion to judge the quantization quality.

## A.3 Ablation Experiments of Scaling Factor Granularity

We designed ablation experiments on the grain of scaling factor. A fine-grain scaling factor can improve the accuracy of quantized models, but it may affect the deployment efficiency on hardware.

Table 3: Ablation Experiments of Feature Maps' Scaling Factor Granularity on Resnet18

| Granularity of feature maps | Tensor-wise | Frequency-wise |
|---|---|---|
| Granularity of filters | Channel-wise | Channel-wise |
| Accuracy | 69.18% | 69.54% |

Form the Table 3, we can find that for feature map, frequency-wise quantization is necessary.

Table 4: Ablation Experiments of Filters' Scaling Factor Granularity on Resnet18

| Granularity of feature maps | Frequency-wise | Frequency-wise | Frequency-wise |
|---|---|---|---|
| Granularity of filters | Channel-wise | Frequency-wise | Frequency-wise+Channel-wise |
| Accuracy | 69.54% | 69.55% | 69.60% |

Form the Table 4, we can find that for filters, channel-wise and frequency-wise yield similar results, both outperforming tensor-wise. Where hardware permits, simultaneous frequency-wise and channel-wise quantization would give the best results.

