# OpenReview forum: "Efficient and Quantization-Friendly Ternary Fourier Convolution Algorithms"
_ICLR.cc/2024/Conference — Submitted to ICLR 2024_

### Official Review · Reviewer_NNvf · 2023-10-29

**Soundness:** 3 good
**Presentation:** 3 good
**Contribution:** 3 good
**Rating:** 6
**Confidence:** 4

**Summary:**

- To address the issue of numerical errors caused by model quantization in the complex domain, the authors present a novel fast convolution algorithm referred to as Ternary Fourier Convolution (TFC) that utilizes ternary matrices for input and weight transformations before multiplication.
- The proposed TFC is derived from the implementation of symbolic arithmetic on the Fourier transform to eliminate the involvement of irrational numbers.
- And, the authors incorporate correction terms to convert ineffective circular convolution results into efficient ones. The proposed method achieves 3.68×, 4.89×, and 4.54× theoretical multiplication complexity reduction for 3×3, 5×5, and 7×7 convolutions, respectively.
- Moreover, the corresponding post-training quantization method requires only a few samples for calibrating network parameters and restoring accuracy without the high cost of retraining. The extensive experiments demonstrate an accuracy drop of less than 0.2% under Int8 quantization for models trained on ImageNet.

**Strengths:**

- (+) The proposed TFC addresses the issue of numerical errors caused by model quantization in a complex domain in a simple way and shows the efficiency in convolutions.
- (+) The method introduces a simple calibrating network parameter to minimize quantization errors and shows comparisons for quantization bits results.

**Weaknesses:**

- (-) It seems to lack qualification results to support Table 2 such as comparisons of feature maps.
- (-) The ablation study on scale factor (e.q. 14) seems to be needed for its effectiveness.
- (-) We need an architecture table to compare one (TFC) with the others regarding layer-wise output size and number of parameters.

**Questions:**

- Please see the above weak points.

**Details Of Ethics Concerns:**

None.

---

> ### Author Response · Authors · 2023-11-19
> **To Reviewer NNvf**
>
> We thank you for your reviews and address your concerns as follows.
>
> **Q1**: It seems to lack qualification results to support Table 2 such as comparisons of feature maps.
>
> **A1**: Since we opt for a classification task on Imagenet dataset in the comparison experiments, the model accuracy serves as an indicator of the algorithm's quality. For example, using our proposed SFC6(7×7, 3×3) algorithm under Int8 arithmetic, the accuracy reduction is to only 0.15%, which substantially outperforms similar fast convolution quantization work.
>
> **Q2**: The ablation study on scale factor (e.q. 14) seems to be need for its effectiveness.
>
> **A2**: We add new ablation experiments on the scaling factor. The detailed results are given in **Table R1-2**.
>
> **Table R1**
>
> | Resnet18(69.76%) | Minmax | Scaling Factor Fine-tune |
> | ---------------- | ------ | ------------------------ |
> | Accuracy         | 67.42  | 69.58                    |
>
> We can see by involving scaling factor fine-tune scheme, the model accuracy is better preserved.
>
> We also add ablation experiments on the grain of scaling factor. A fine-grain scaling factor can improve the accuracy of quantized models, but it may affect the deployment efficiency on hardware. For activation, frequency-wise quantization is necessary. For filters, channel-wise and frequency-wise yield similar results, both outperforming tensor-wise. Where hardware permits, simultaneous frequency-wise and channel-wise quantization would give the best results.
>
> **Table R2**
>
> | scaling factor grain of Activations | Tensor-wise  | Frequency-wise | Frequency-wise | Frequency-wise              |
> | ----------------------------------- | ------------ | -------------- | -------------- | --------------------------- |
> | scaling factor grain of filters     | Channel-wise | Channel-wise   | Frequency-wise | Frequency-wise+Channel-wise |
> | Accuracy                            | 69.18        | 69.54          | 69.55          | 69.58                       |
>
> **Q3**: We need an architecture table to compare one (TFC) with the others regarding layer-wise output size and number of parameters.
>
> **A3**: Applying our method or a fast convolution algorithm does not increase the number of parameters in the model or change the size of the layer output. The parameters of the transformation matrix in our algorithms are often incorporated as fixed values in the program or designed for dedicated domain accelerators, rather than being stored as additional parameters. When combining our method with quantization, the storage of the quantized scaling factor is required, but this demand is minimal compared to the number of original parameters in the network.

---

> > ### Comment · Reviewer_NNvf · 2023-11-21
> >
> > Thank you for your detailed explanations and extensive experiments.  I will keep my score. It needs to include more ablation studies on feature maps and training/test time to show its effectiveness.

---

> ### Author Response · Authors · 2023-11-21
> **To Reviewer NNvf Part II**
>
> Thank you for your timely and helpful feedback. We have made revision to address your concerns in the latest revised version. Let we address your concerns as follows.
>
>
>
> 1. **Absolution studies on feature map.**
>
>  In the **Appendix A.2**, we plot the layer-wise Mean Squared Error (MSE) error of feature maps as the qualification metric to compared different algorithms, using AdaQuant[1] as the fine-tune method. Some of its contents are shown in Table R3:
>
> **Table R3. MSE error of feature maps for different algorithms**
>
> | Resnet18,conv4.1.2    | Standard convolution |        | Winograd (4×4, 3×3) |      | SFC6 (7×7, 3×3) |        |
> | --------------------- | -------------------- | ------ | ------------------- | ---- | --------------- | ------ |
> | is fine-tune          | No                   | Yes    | No                  | Yes  | No              |        |
> | MSE error to FP model | 2.7e-3               | 2.8e-4 | 1.3                 | 0.36 | 2.3e-2          | 6.8e-3 |
>
> We can see that, in our proposed SFC algorithm, **the MSE loss of feature maps is significantly lower than that in the Winograd algorithm**. However, it is worth noting that, when using the scaling-factor fine-tune method, the MSE loss of feature maps in the SFC algorithm is smaller than that in the standard convolution, while the accuracy of the quantized model does not improve. It enlightens us that the MSE error of the feature map is not the sole criterion to judge the quantization quality.
>
> 2. **Absolution studies on training time.**
>
> We compared our PTQ method and the QAT[2] method on VGG-16 networks under CIFAR-10 dataset to prove the effectiveness of our method.
>
> However, we observed that, when employing the SFC4(44,33) algorithm with sufficient quantization-aware training time, the final accuracy can surpass that of the floating-point model. We suspect that this improvement is attributed to the fact that the SFC algorithm is based on the Fourier transform, and the processing properties for images endowed by the Fourier transform (referred to as the **Frequency Principle[5]**) enhance the model's performance after extensive training. As a result, we provide two quantization-aware training times: one for recovering the accuracy of the FP32 model and another for converging to the highest accuracy, serving as a reference.
>
> **Table R4. Training time**
>
> | VGG16 (92.6%) | QAT[2] 8bit  |              | PTQ (ours) 8bit |
> | ------------- | ------------ | ------------ | --------------- |
> | Time          | 22min(92.6%) | 39min(93.4%) | 17s(92.6%)      |
>
> Our PTQ method accelerates the training time for recovering model accuracy by a factor of 78×.
>
>
>
> 3. **Absolution studies on testing(inference) time.**
>
> There has been considerable work on deploying & optimizing Winograd algorithm on CPUs[3] and GPUs[4]. As our algorithm can be computed in the same computational flow as the Winograd, its deployment has no serious technical problems.
>
> We implement our proposed symbolic Fourier convolution algorithm on our laptop GPU to validate that its can provides real-world inference speedups, substantiating not only theoretical computational reductions. For comparison, we evaluated our method against the CUDA Deep Neural Network library (cuDNN)[6] in the context of the 3×3 convolution layer for inference. Because of the time limitation, we were unable to implement the algorithm across entire networks or in Int8 format. As we can see, our algorithms demonstrates a 1.75x speedup compared to cuDNN.
>
> **Table R5. Inference Convlayer with BatchSize=1, KernelSize=3×3，FeatureMap=28×28，Channel=128×128 and fp32 format on Nvidia GTX 1650 Laptop GPU**
>
> |      | cuDNN | im2col | SFC4(4×4, 3×3) |
> | ---- | :---: | ------ | -------------- |
> | time | 128us | 194us  | **73us**           |
>
>
>
> [1] Hubara, Itay, et al. "Improving post training neural quantization: Layer-wise calibration and integer programming." *arXiv preprint arXiv:2006.10518* (2020).
>
> [2] Andri, Renzo, et al. "Going further with winograd convolutions: Tap-wise quantization for efficient inference on 4x4 tiles." *2022 55th IEEE/ACM International Symposium on Microarchitecture (MICRO)*. IEEE, 2022.
>
> [3] Maji, Partha, et al. *Efficient Winograd or Cook-Toom Convolution Kernel Implementation on Widely Used Mobile CPUs*. IEEE, 2019
>
> [4] Jia, Liancheng, et al. “Enabling Efficient Fast Convolution Algorithms on GPUs via MegaKernels.” *IEEE Transactions on Computers*, 2020, pp. 1–1
>
> [5] Xu, Z., et al. “Overview Frequency Principle/Spectral Bias in Deep Learning.” *arXiv.Org*, 2022.
>
> [6] Chetlur, Sharan, et al. *cuDNN: Efficient Primitives for Deep Learning*. 2014.

---

### Official Review · Reviewer_tjrq · 2023-10-31

**Soundness:** 3 good
**Presentation:** 3 good
**Contribution:** 2 fair
**Rating:** 3
**Confidence:** 4

**Summary:**

This paper explores techniques for accelerating fast convolution algorithms based on Fourier transform through quantization. To mitigate the impact of quantization error, the paper utilized a ternary transformation matrix derived from symbolic arithmetic applied to the Fourier transform. Furthermore, the paper combines post-training quantization (PTQ) approach to enable INT8 quantization. The methods achieve a reduction on theoretical computation complexity while showing negligible accuracy loss on ResNet ImageNet benchmarks.

**Strengths:**

1). The paper is well written. The algorithm is clearly explained and easy to follow.

2). The results show clear improvement on both the computation complexity reduction, while preserving the model accuracy at INT8.

**Weaknesses:**

1). The use of “Ternary” in the title is misleading, as ternary refers to an intermediate transform matrix; the actual precision is INT8.

2). One of the primary concerns is the speedup achieved through current approach. The paper only provides theoretical multiplication complexity reduction, which may not necessarily translate to real-world speedup. Given the complexity of the algorithm and the potential overhead, it is important to provide empirical measurements of computation efficiency.

3). PTQ method is a well-established technique.

4). The evaluation is only performed on ResNet models, which, while important, are somewhat outdated. It would be benificail to evaluate the methods across a more extensive range of CNN models, particularly the compact ones like EfficientNet and MobileNet.

5). Today, numerous techniques exist for accelerating CNN models, including the design of compact architecture, sparsity, distillation, and quantization. For quantization alone, it is possible to reduce precision to sub-4 bit level while preserving model accuracy. The paper only compares its approach to other fast convolution algorithms, such as Winograde, and while it demonstrates improvement, it is challenging to evaluate the significance of this approach compared to toher alternatives.

**Questions:**

In addition to weaknesses above,

1). How will this method be used for filters with smaller sizes, such as 1x1 and depth-wise conv layers?

2). Figure 3, the blue line for standard convolution does not seem to use SOTA PTQ techniques, which can effectively close the accuracy gap for INT4 (such as ref. 1 and 2).

3). Typo? Page 2, contribution 1 paragraph, x, x, and 7x7.

---

> ### Author Response · Authors · 2023-11-19
> **To Reviewer tjrq Part I**
>
> Thank you for pointing out many important issues. We have made revisions to address your concerns in the revised version. The revision can be summarized as follows:
>
> 1. We have revised the name of our proposed algorithm to Symbolic Fourier Convolution (SFC), replacing the prior name of Ternary Fourier Convolution (TFC).
>
> 2. We have involved Bit Operations(BOPs)[1] [2] [3] as a alternative metric to precisely quantify computation costs/acceleration ratio.
>
> 3. We have involved a sota PTQ baseline AdaQuant[4], this method improve the result of bsl and our algorithms under Int4 quantization.
>
> 4. We have corrected the typos in the revised version.
>
> And then let we address your concerns as follows.
>
> **Q1**:  The use of “Ternary” in the title is misleading, as ternary refers to an intermediate transform matrix; the actual precision is INT8.
>
> **A1**: We apologize for the confuse in our initial preparation. We have updated the name of our proposed algorithm to **Symbolic Fourier Convolution (SFC)**, replacing the previous designation of Ternary Fourier Convolution (TFC). The new name more accurately reflects the underlying concept of our algorithm. The earlier name could potentially be misleading, suggesting the use of three-valued computation throughout the entire convolution process.
>
>
>
> **Q2**:  One of the primary concerns is the speedup achieved through current approach. The paper only provides theoretical multiplication complexity reduction, which may not necessarily translate to real-world speedup. Given the complexity of the algorithm and the potential overhead, it is important to provide empirical measurements of computation efficiency.
>
> **A2**: Thanks you for your suggestion. **First, we implement our proposed symbolic Fourier convolution algorithm on our Laptop GPU to validate that its can provides real-world speedups**, substantiating not only theoretical computational reductions. For comparison, we evaluated our method against the CUDA Deep Neural Network library (cuDNN) in the context of the 3×3 convolution layer for inference. Because of the time limitation, we were unable to implement the algorithm across entire networks in Int8 format.  As we can see, our algorithms demonstrates a 1.75x speedup compared to cuDNN.
>
> **Table R1. Inference fp32 Conv with BatchSize=1, KSize=3×3，FMSize=28×28，C=128×128 on Nvidia GTX 1650 Laptop GPU**
>
> |      | cuDNN | im2col | SFC  |
> | ---- | :---: | ------ | ---- |
> | time | 128us | 194us  | 73us |
>
> Furthermore, we opt for **Bit-Operations(BOPs)** as a fine metric to precisely quantify computation costs.  This metric comprehensively considers factors such as **bit-width,  operations number, and the varying hardware costs of addition and multiplication**. And it is widely used in various model compressing fields, including  Neural Architecture Search(NAS)[6], quantization[5] [7] and pruning [5] research.  BOPs for integer addition are defined as $BOPs = b \cdot ADDs$, where '$b$' represents the addition bit-width. Similarly, BOPs for integer multiply-and-accumulate are defined as $BOPs = b1 \cdot b2 \cdot MACs$, where '$b1$' and '$b2$' represent the bit-widths of the two operators, respectively.
>
> For element-wise multiplication, our method consumes $MACs=Tnums\cdot TransSize\cdot TransSize\cdot IC\cdot OC$ when deploying a dense convolution layer, where $ Tnums = ceil(H/TileSize)\cdot ceil(W/TileSize)$. And the input transformation consumes $ADDs = Tnum\cdot ((TileSize+KernelSize)\cdot TransSize)(TileSize+KernelSize+TransSize)\cdot IC$,  the output transformations consumes $ADDs = Tnum\cdot (TileSize\cdot TransSize)(TileSize+TransSize)\cdot OC$.
>
> When the arithmetic bit-width=8, we can find that $BOPs_{ElementWiseMult} >> BOS_{OutTrans} \approx BOS_{InTrans}$, which is why many works based on fast convolution primarily considers only element-wise multiplication numbers when calculating the speedup ratio.
>
> The BOPs of different methods are shown in Table R2. Compared to the state-of-the-art PTQ method AdaQuant[1], SFC method can **significantly reduce the MACs while maintaining narrow bit-width in multiplication**, thus obtaining lower BOPs. In comparison with Winograd-based work[2], our algorithm **does not need to expand the intermediate arithmetic bit-width to maintain the model accuracy**, and thus the BOPs are also lower.
>
> **Table R2. BOPs of ResNet18(69.76%)**
>
> |           | AdaQuant(PTQ)[1] |       | Wino(4×4, 3×3)+QAT[2] | SFC6(7×7, 3×3)+PTQ | SFC4(4×4, 3×3)+PTQ | SFC4(4, 3)+PTQ |
> | --------- | ---------------- | ----- | --------------------- | ------------------ | ------------------ | -------------- |
> | Bit-width | 8                | 4     | 8/10                  | 8                  | 8/6                | 8/4            |
> | Accuracy  | 69.73            | 63.47 | 69.41                 | 69.58              | 69.04              | 63.02          |
> | BOPs      | 96.2G            | 24.1G | 44.3G                 | **34.3G**          | **22.8G**          | **15.8G**      |

---

> ### Author Response · Authors · 2023-11-19
> **To Reviewer tjrq Part II**
>
> **Q3**:  PTQ method is a well-established technique.
>
> **A3**: While the PTQ method is a well-established technique, previous work could not perfectly combine it with fast convolutional algorithms to achieve further speedup. For example, the method in ref [3] results in a 2.2%  accuracy reduction for Resnet18 under Int8 PTQ, and in ref [4], there was a 4.6%  accuracy reduction in Resnet18 under Int6 PTQ.  In contrast, our method can be effectively combined with the PTQ with only 0.1% accuracy loss under Int8 and 0.7% under Int6 (Table R2).
>
>
>
> **Q4**: The evaluation is only performed on ResNet models, which, while important, are somewhat outdated. It would be benificail to evaluate the methods across a more extensive range of CNN models, particularly the compact ones like EfficientNet and MobileNet.
>
> **A4**：In this paper, we done experiments on ResNets and compared with other works. Here, we add new experiments on VGG16, EfficientNetB0 and Mobilenetv2 under the Cifar10 datasets. The results are in alignment with our expectations, and all models consistently maintained accuracy.
>
> **Table R3. Accuracy on VGG16, Mobilnetv2 and EfficientnetB0**
>
> | Model      | VGG16 (92.64%)   |                  | Mobilnetv2 (94.43%) |                  | EfficientnetB0 (94.73%) |                  |
> | ---------- | ---------------- | ---------------- | ------------------- | ---------------- | ----------------------- | ---------------- |
> | Algorithms | TFC4(44,33)+Int8 | TFC6(66,33)+Int8 | TFC4(44,33)+Int8    | TFC6(66,33)+Int8 | TFC4(44,33)+Int8        | TFC6(66,33)+Int8 |
> | Accuracy   | 92.79%           | 92.61%           | 94.38%              | 94.37%           | 94.67%                  | 94.75%           |
>
>
>
> **Q5:** Today, numerous techniques exist for accelerating CNN models, including the design of compact architecture, sparsity, distillation, and quantization. For quantization alone, it is possible to reduce precision to sub-4 bit level while preserving model accuracy. The paper only compares its approach to other fast convolution algorithms, such as Winograd, and while it demonstrates improvement, it is challenging to evaluate the significance of this approach compared to other alternatives.
>
> **A5:** **The motivation of this paper is to enhance the compatibility of fast convolution algorithms with quantization methods, aiming to achieve improved compression performance by combining them**. Fast convolutional algorithms have many advantages, including their **suitability across various hardware platforms**, including CPUs[8], GPUs[9], domain specific accelerators[2]. Other acceleration methods like extreme low-bit quantization and sparsity are more dependent on the support of the hardware architecture for their respective function. For example, Nvidia Turing GPUs can only support specific quantization formats (Int8 and Int16) and sparsity templates (4:2 sparsity).
>
> Discussion in terms of quantization alone: Although achieving sub-4bit quantization and maintaining model accuracy is possible through Quantization-aware Training (QAT) [10], it requires the entire dataset and significant training resources.  In our work, we leverage Post-training quantization (PTQ), which is more lightweight, requiring only a small dataset and estimating parameters in a few minutes. Compared to the sota PTQ alone method AdaQuant[1], our method can achieve **much higher accuracy under similar BOPs or much lower BOPs under similar accuracy**(Table R4). Finally, we believe that our  SFC algorithm can also achieve sub 4-bit quantization via QAT, although we have not attempted it yet.
>
>  **Table R4. BOPs of ResNet18(69.76%)**
>
> |           | AdaQuant(PTQ)[1] | SFC4(4×4, 3×3)+PTQ | SFC4(4, 3)+PTQ |
> | --------- | ---------------- | ------------------ | -------------- |
> | Bit-width | 4                | 8/6                | 8/4            |
> | Accuracy  | 63.47            | **69.04**          | 63.02          |
> | BOPs      | 24.1G            | 22.8G              | **15.8G**      |
>
>
>
> Discussion in terms of sparsity: The irregular data flow resulting from sparsity poses a significant challenge to hardware architecture design, often leading to hardware inefficiency. Unlike PTQ methods, sparse methods, to maintain accuracy, have high training costs. Furthermore, sparse methods are not incompatible with fast convolutional algorithms; **a previous study [11] explores sparsity in the Winograd domain**. Therefore, we believe that our SFC algorithm can be effectively combined with sparsity methods in the transformation domain.
>
> Discussion in terms of distillation: We think our algorithm is not in conflict with knowledge distillation. Our method provide a fast algorithm for convolution and it is also friendly for quantization. And knowledge distillation is to train a smaller, student model to reproduce the behavior of a larger, teacher model. We can view them as distinct  stages in the model compression pipeline.

---

> ### Author Response · Authors · 2023-11-19
> **To Reviewer tjrq Part III**
>
> **Q6**: How will this method be used for filters with smaller sizes, such as 1x1 and depth-wise conv layers?
>
> **A6**: Our method can accelerate 3×3 and lager size convolutions very efficiently (the only fast convolution algorithm to our knowledge that can accelerate 3×3 convolutions by more than 3× at Int8 without loss of accuracy), but it does not work for 1×1 convolutions. However, our algorithm may affect the application value of the 1×1 convolution. In many compact architectures, 1×1 convolutions are designed to reduce calculations, but can not extract features in the spatial dimension like 3×3. When the overhead of 3×3 convolution is greatly reduced (our method can reduce it by 3.68×), the significance of this design purpose may not be as significant as before.
>
> Our method can work for depth-wise convolution. However, according to the roof-line model, the inference bottleneck for depth-wise convolution is not in computation but in data transmission(low activation reuse on channel dimension). Thus reducing its computational complexity may not increase its inference speed. Data compression techniques for activation and weights may be more significant for accelerating depth-wise convolution.
>
>
> **Q7**:  Figure 3, the blue line for standard convolution does not seem to use SOTA PTQ techniques, which can effectively close the accuracy gap for INT4 (such as ref. 1 and 2).
>
> **A7**:  Thanks for your suggestion, in our revised manuscript we involved a sota PTQ baseline **AdaQuant[1]**, this method improve the result of bsl and our algorithms under 4-bit quantization. **However, for the Winograd convolution, there are convergence problems and the results are even worse**. This is the why the way recent quantization works for Winograd unify utilized method based on gradients backward propagation[2] [3] [4].
>
> **Table R5. Partial results of Figure 3 in the revised manuscript**
>
> |           | Wino(44,33)        |              | standard convolution |          | SFC(77,33)         |          | SFC(4, 3) |
> | --------- | ------------------ | ------------ | -------------------- | -------- | ------------------ | -------- | --------- |
> | method    | gradients backward | AdaQuant     | gradients backward   | AdaQuant | gradients backward | AdaQuant | AdaQuant  |
> | bit-width | 8                  | 8            | 4                    | 4        | 4                  | 4        | 4         |
> | accuracy  | 67.32              | **8.586↓↓↓** | 60.52                | 63.47↑   | 53.82              | 55.84↑   | 63.02     |
>
>
>
> **Q8**: Typo? Page 2, contribution 1 paragraph, x, x, and 7x7.
>
> **A8**: Thank you tor your careful reading. We apologize for our oversights. We have corrected these typos in our revised manuscript .
>
>
>
> [1] Hubara, Itay, et al. "Improving post training neural quantization: Layer-wise calibration and integer programming." *arXiv preprint arXiv:2006.10518* (2020).
>
> [2] Andri, Renzo, et al. "Going further with winograd convolutions: Tap-wise quantization for efficient inference on 4x4 tiles." *2022 55th IEEE/ACM International Symposium on Microarchitecture (MICRO)*. IEEE, 2022.
>
> [3] Chikin, Vladimir, and Vladimir Kryzhanovskiy. "Channel balancing for accurate quantization of winograd convolutions." *Proceedings of the IEEE/CVF Conference on Computer Vision and Pattern Recognition*. 2022.
>
> [4] Tianqi, Chen, et al. "Towards Efficient and Accurate Winograd Convolution via Full Quantization." *Thirty-seventh Conference on Neural Information Processing Systems*. 2023.
>
> [5] Wang, Ying, Yadong Lu, and Tijmen Blankevoort. "Differentiable joint pruning and quantization for hardware efficiency." *European Conference on Computer Vision*. Cham: Springer International Publishing, 2020.
>
> [6] Guo, Zichao, et al. "Single path one-shot neural architecture search with uniform sampling." *Computer Vision–ECCV 2020: 16th European Conference, Glasgow, UK, August 23–28, 2020, Proceedings, Part XVI 16*. Springer International Publishing, 2020.
>
> [7] Liu, Zechun, et al. "Reactnet: Towards precise binary neural network with generalized activation functions." *Computer Vision–ECCV 2020: 16th European Conference, Glasgow, UK, August 23–28, 2020, Proceedings, Part XIV 16*. Springer International Publishing, 2020.
>
> [8] Maji, Partha, et al. *Efficient Winograd or Cook-Toom Convolution Kernel Implementation on Widely Used Mobile CPUs*. IEEE, 2019
>
> [9] Jia, Liancheng, et al. “Enabling Efficient Fast Convolution Algorithms on GPUs via MegaKernels.” *IEEE Transactions on Computers*, 2020, pp. 1–1
>
> [10] Rastegari, Mohammad, et al. "Xnor-net: Imagenet classification using binary convolutional neural networks." *European conference on computer vision*. Cham: Springer International Publishing, 2016.
>
> [11] Liu, Xingyu, et al. “Efficient Sparse-Winograd Convolutional Neural Networks.” *International Conference on Learning Representations*, 2018.

---

### Official Review · Reviewer_uiWW · 2023-10-31

**Soundness:** 2 fair
**Presentation:** 1 poor
**Contribution:** 2 fair
**Rating:** 5
**Confidence:** 2

**Summary:**

The paper introduces a new fast convolution algorithm that uses ternary matrices for transformations, minimizing these quantization errors. This technique is based on symbolic arithmetic applied to the Fourier transform, avoiding irrational numbers, and includes correction terms to improve the convolution results' efficiency. A novel post-training quantization method is also proposed, which calibrates network parameters using only a few samples, thus avoiding the need for expensive retraining. The proposed algorithms significantly reduce multiplication complexity—up to 4.89× for common convolution sizes—and demonstrate an impressively low accuracy drop of less than 0.2% on ImageNet models under Int8 quantization. This performance surpasses competing methods, offering a more efficient and accurate approach for CNN deployment in resource-constrained environments.

**Strengths:**

1. using symbolic operation instead of numerical computation is very attractive. The demonstration of the proposed idea is very solid and sound.
2. I really appreciate the quantization method based on the frequency. The observation that a relation between the energy distribution and the frequency channel coordinates is very promising.
2. The evaluation result is very significant, about a 5x reduction in multiplicative complexity compared with other works.

**Weaknesses:**

1. The writing of this paper is very hard to follow. Many typos are in the paper, such as some numbers are missing in the first contribution of section Introduction " x x ", and the bottom line is missing in Table 2.
2. The motivation for this paper is not clear. Since many compression works like extreme low-bit quantization, pruning, and low-rank decomposition are proposed to accelerate the convolution layers, the motivation using Fourier transformation is not clear.
3. The evaluation is not sufficient. The metric in evaluation is multiplicative complexity, however, the compression ratio and real-time acceleration performance are missing.

**Questions:**

1. It would be better to provide the real-time latency to show the acceleration performance on hardware platforms like GPU/FPGA/CPU. The theoretical reduction in the multiplication operation.
2. Can this method be used in other layers? For example, FC can be seen as a special CNN layer with a kernel size being 1, and what if the proposed method is applied to FC layers?

---

> ### Author Response · Authors · 2023-11-19
> **To Reviewer uiWW Part I**
>
> Thank you for pointing out many important issues. We have made revisions to address your concerns in the revised version. The revision can be summarized as follows:
>
> 1. We have corrected the typos in the revised version.
> 2. We have strengthened the expression of motivation in the abstract and introduction.
> 3. We have involved Bit Operations(BOPs)[1] [2] [3] as a alternative metric to precisely quantify computation costs/acceleration ratio.
>
> And then let we address your concerns as follows.
>
> **Q1**: The writing of this paper is very hard to follow. Many typos are in the paper, such as some numbers are missing in the first contribution of section Introduction " x x ", and the bottom line is missing in Table 2.
>
> **A1**: Thank you tor your careful reading of our paper. We have corrected these typos in our latest manuscript .
>
>
> **Q2**: The motivation for this paper is not clear. Since many compression works like extreme low-bit quantization, pruning, and low-rank decomposition are proposed to accelerate the convolution layers, the motivation using Fourier transformation is not clear.
>
> **A2**: The motivation of this paper is to **enhance the compatibility of fast convolution algorithms with quantization methods, aiming to achieve improved computation efficiency by combining them**. Thanks to your suggestion, we have strengthened the expression of motivation in the revised manuscript. Experiment section shows that our method is far ahead of similar work in terms of accuracy, quantization bit-width, and speedup ratio.
>
> Compared with other compression methods, fast convolution algorithms have many advantages, including their **suitability across various hardware platforms**, including CPUs[8], GPUs[9], domain specific accelerators[2]. Other acceleration methods like extreme low-bit quantization and pruning are more dependent on the support of the hardware architecture for their respective function. For example, Nvidia Turing GPUs can only support specific quantization formats (Int8 and Int16) and pruning templates (4:2 sparsity).
>
> Discussion in terms of quantization alone: Although achieving sub-4bit quantization and maintaining model accuracy is possible through Quantization-aware Training (QAT) [10], it requires the entire dataset and significant training resources.  In our work, we leverage Post-training quantization (PTQ), which is more lightweight, requiring only a small dataset and estimating parameters in a few minutes. Compared to the sota PTQ alone method AdaQuant[1], our method can **achieve much higher accuracy under similar BOPs** (Bit Operations, which we will detail this metric in **A3** section) or **much lower BOPs under similar accuracy**(Table R1). Finally, we believe that our SFC algorithm can also achieve sub 4-bit quantization via QAT, although we have not attempted it yet.
>
>  **Table R1. BOPs of ResNet18(69.76%)**
>
> |           | AdaQuant(PTQ)[1] | SFC4(4×4, 3×3)+PTQ | SFC4(4, 3)+PTQ |
> | --------- | ---------------- | ------------------ | -------------- |
> | Bit-width | 4                | 8/6                | 8/4            |
> | Accuracy  | 63.47            | **69.04**          | 63.02          |
> | BOPs      | 24.1G            | 22.8G              | **15.8G**      |
>
>
>
> Discussion in terms of pruning: The irregular data flow resulting from pruning poses a significant challenge to hardware architecture design, often leading to hardware inefficiency. Unlike PTQ methods, pruning methods, to maintain accuracy, have high training costs. Furthermore, pruning methods are not incompatible with fast convolutional algorithms; **a previous study [4] explores sparsity in the Winograd domain**. Therefore, we believe that our SFC algorithm can be effectively combined with pruning methods in the transformation domain.
>
> Discussion in terms of low-rank decomposition: Compared to low-rank decomposition, which provides only **approximate computations**, fast convolution algorithms provide **mathematically equivalent computations**, once numerical errors are within model tolerance, the loss of accuracy is less.

---

> ### Author Response · Authors · 2023-11-19
> **To Reviewer uiWW Part II**
>
> **Q3**：The evaluation is not sufficient. The metric in evaluation is multiplicative complexity, however, the compression ratio and real-time acceleration performance are missing.
>
> **A3**：Thanks you for your suggestion, we opt for **Bit-Operations(BOPs)** in our revised manuscript as a fine metric to precisely quantify computation costs.  This metric comprehensively considers factors such as **bit-width, the number of operations, and the varying hardware costs of addition and multiplication**. And it is widely used in various model compressing fields, including  Neural Architecture Search(NAS)[6], quantization[5] [7] and pruning [5] research.  BOPs for integer addition are defined as $BOPs = b \cdot ADDs$, where '$b$' represents the addition bit-width. Similarly, BOPs for integer multiply-and-accumulate are defined as $BOPs = b1 \cdot b2 \cdot MACs$, where '$b1$' and '$b2$' represent the bit-widths of the two operators, respectively.
>
> For element-wise multiplication, our method consumes $MACs=Tnums\cdot TransSize\cdot TransSize\cdot IC\cdot OC$ when deploying a dense convolution layer, where $ Tnums = ceil(H/TileSize)\cdot ceil(W/TileSize)$. And the input transformation consumes $ADDs = Tnum\cdot ((TileSize+KernelSize)\cdot TransSize)(TileSize+KernelSize+TransSize)\cdot IC$,  the output transformations consumes $ADDs = Tnum\cdot (TileSize\cdot TransSize)(TileSize+TransSize)\cdot OC$.
>
> When the arithmetic bit-width=8, we can find that $BOPs_{ElementWiseMult} >> BOS_{OutTrans} \approx BOS_{InTrans}$, **which is why many works based on fast convolution primarily considers element-wise multiplication complexity when calculating the speedup ratio.**
>
> The BOPs of different methods are shown in Table R2. Compared to the state-of-the-art PTQ method AdaQuant[1], our SFC method can **significantly reduce the MACs while maintaining narrow bit-width in multiplication**, thus obtaining lower BOPs. In comparison with Winograd-based work[2], our algorithm does **not need to expand the intermediate arithmetic bit-width to maintain the model accuracy**, and thus the BOPs are also lower.
>
> **Table R2. BOPs of ResNet18(69.76%)**
>
> |           | AdaQuant(PTQ)[1] |       | Wino(4×4, 3×3)+QAT[2] | SFC6(7×7, 3×3)+PTQ | SFC4(4×4, 3×3)+PTQ | SFC4(4, 3)+PTQ |
> | --------- | ---------------- | ----- | --------------------- | ------------------ | ------------------ | -------------- |
> | Bit-width | 8                | 4     | 8/10                  | 8                  | 8/6                | 8/4            |
> | Accuracy  | 69.73            | 63.47 | 69.41                 | 69.58              | 69.04              | 63.02          |
> | BOPs      | 96.2G            | 24.1G | 44.3G                 | **34.3G**          | **22.8G**          | **15.8G**      |
>
>
> **Q4**. It would be better to provide the real-time latency to show the acceleration performance on hardware platforms like GPU/FPGA/CPU. The theoretical reduction in the multiplication operation.
>
> **A4**：There has been considerable works on deploying & optimizing Winograd algorithm on CPUs[8] and GPUs[9]. As our algorithm can be computed in the same computational flow as the Winograd, we think its deployment has no serious technical problems.
>
> We implement our proposed symbolic Fourier convolution algorithm on our laptop GPU to validate that its can provides real-world speedups, substantiating not only theoretical computational reductions. For comparison, we evaluated our method against the CUDA Deep Neural Network library (cuDNN) in the context of the 3×3 convolution layer for inference. Because of the time limitation, we were unable to implement the entire networks in Int8 format. As we can see, **our algorithms demonstrates a 1.75x speedup compared to cuDNN**.
>
> **Table R3. Inference Convlayer with BatchSize=1, KernelSize=3×3，FeatureMap=28×28，Channel=128×128 and fp32 format on Nvidia GTX 1650 Laptop GPU**
>
> |      | cuDNN | im2col | SFC4(44, 33)  |
> | ---- | :---: | ------ | ---- |
> | time | 128us | 194us  | **73us** |
>
> We are also working on FPGA-based CNN accelerator utilizing our proposed algorithm. Welcome to follow our hardware work in the future.

---

> ### Author Response · Authors · 2023-11-19
> **To Reviewer uiWW Part III**
>
> **Q5**: Can this method be used in other layers? For example, FC can be seen as a special CNN layer with a kernel size being 1, and what if the proposed method is applied to FC layers?
>
> **A5**: Our method can accelerate 3×3 and lager size convolutions very efficiently (the only fast convolution algorithm to our knowledge that can accelerate 3×3 convolutions by more than 3× at 8bit with negligible accuracy loss), but it does not work for FC layers. However, in a significant number of CNNs, 3×3 convolution takes up a heavy computational load, much more than FC.
>
>
>
> [1] Hubara, Itay, et al. "Improving post training neural quantization: Layer-wise calibration and integer programming." *arXiv preprint arXiv:2006.10518* (2020).
>
> [2] Andri, Renzo, et al. "Going further with winograd convolutions: Tap-wise quantization for efficient inference on 4x4 tiles." *2022 55th IEEE/ACM International Symposium on Microarchitecture (MICRO)*. IEEE, 2022.
>
> [3] Chetlur, Sharan, et al. *cuDNN: Efficient Primitives for Deep Learning*. 2014.
>
> [4] Liu, Xingyu, et al. “Efficient Sparse-Winograd Convolutional Neural Networks.” *International Conference on Learning Representations*, 2018.
>
> [5] Wang, Ying, Yadong Lu, and Tijmen Blankevoort. "Differentiable joint pruning and quantization for hardware efficiency." *European Conference on Computer Vision*. Cham: Springer International Publishing, 2020.
>
> [6] Guo, Zichao, et al. "Single path one-shot neural architecture search with uniform sampling." *Computer Vision–ECCV 2020: 16th European Conference, Glasgow, UK, August 23–28, 2020, Proceedings, Part XVI 16*. Springer International Publishing, 2020.
>
> [7] Liu, Zechun, et al. "Reactnet: Towards precise binary neural network with generalized activation functions." *Computer Vision–ECCV 2020: 16th European Conference, Glasgow, UK, August 23–28, 2020, Proceedings, Part XIV 16*. Springer International Publishing, 2020.
>
> [8] Maji, Partha, et al. *Efficient Winograd or Cook-Toom Convolution Kernel Implementation on Widely Used Mobile CPUs*. IEEE, 2019
>
> [9] Jia, Liancheng, et al. “Enabling Efficient Fast Convolution Algorithms on GPUs via MegaKernels.” *IEEE Transactions on Computers*, 2020, pp. 1–1
>
> [10] Rastegari, Mohammad, et al. "Xnor-net: Imagenet classification using binary convolutional neural networks." *European conference on computer vision*. Cham: Springer International Publishing, 2016.

---

### Author Response · Authors · 2023-11-19
**General Response**

Thank you reviewers for the insightful feedback and constructive comments. In response to the reviewers' comments, we have revised the manuscript to enhance the overall quality of our work. The main revisions are summarized as follows:

1. We have corrected the typos in the revised manuscript. We sincerely apologize for any oversights in our initial preparation.

2. We have involved Bit Operations(BOPs)[1] [2] [3] as a alternative metric to precisely quantify computation costs/acceleration ratio.

3. We include a recently received related work from NIPS 2024, which has just been released, in our experimental comparison.

4. We have revised the name of our proposed algorithm to Symbolic Fourier Convolution (SFC), replacing the prior name of Ternary Fourier Convolution (TFC).

5. We have involved a sota PTQ baseline AdaQuant[4], this method improve the result of bsl and our algorithms under Int4 quantization.

6. We have added experiment for 1D fast algorithms in Figure 3, and we found that SFC4(4, 3) can provide1.52× speed up compared to standard convolution under Int4, with similar accuracy. We suggested that using SFA 1D algorithms under Int4 and applying SFA 2D algorithms under Int8/Int6.

7. We have added ablation experiments of scaling factor granularity and MSE comparison of feature maps for different algorithms in Appendix.


[1] Wang, Ying, Yadong Lu, and Tijmen Blankevoort. "Differentiable joint pruning and quantization for hardware efficiency." *European Conference on Computer Vision*. Cham: Springer International Publishing, 2020.

[2] Guo, Zichao, et al. "Single path one-shot neural architecture search with uniform sampling." *Computer Vision–ECCV 2020: 16th European Conference, Glasgow, UK, August 23–28, 2020, Proceedings, Part XVI 16*. Springer International Publishing, 2020.

[3] Liu, Zechun, et al. "Reactnet: Towards precise binary neural network with generalized activation functions." *Computer Vision–ECCV 2020: 16th European Conference, Glasgow, UK, August 23–28, 2020, Proceedings, Part XIV 16*. Springer International Publishing, 2020.

[4] Hubara, Itay, et al. "Improving post training neural quantization: Layer-wise calibration and integer programming." *arXiv preprint arXiv:2006.10518* (2020).

---

### Meta-Review · Area_Chair_uE4s · 2023-12-10

**Metareview:**

This paper explores a new fast convolution algorithm based on Fourier transform through quantization. To mitigate the impact of quantization error, the paper utilized a ternary transformation matrix for input and weight transformations before multiplication. Furthermore, the paper combines post-training quantization (PTQ) approach to enable INT8 quantization. The method significantly reduces multiplication complexity (up to 4.89× for common convolution sizes) while showing negligible accuracy loss on ResNet ImageNet benchmarks.

While presenting a promising idea and positive results, the reviewers suggest comparing to more up-to-date baselines and strengthen the paper with more ablation studies. The paper is much stronger post-rebuttal, and we look forward to see a future submission after addressing those suggestions.

**Justification For Why Not Higher Score:**

It's a very promising idea with positive evidence. However the writing and presentation can be improved. Experiments can be enhanced (with up-to-date benchmarks) to show more supportive signals.

**Justification For Why Not Lower Score:**

N/A

---

### Decision · Program_Chairs · 2024-01-16

Reject